# Factors Influencing Smoking among Multicultural Adolescents

**DOI:** 10.3390/ijerph191811219

**Published:** 2022-09-07

**Authors:** Jin-Hee Park, Mi-Jin Kim, Sung-Eun Kim

**Affiliations:** 1Department of Nursing, Changshin University, Changwon 51352, Korea; 2Department of Nursing, Daegu Haany University, Gyeongsan-si 38610, Korea; 3Department of Nursing, Daedong College, Busan 46270, Korea

**Keywords:** adolescent, smoking, cultural diversity, alcohol

## Abstract

Since an increasing number of multicultural adolescents have started smoking in Korean society, it is necessary to check the smoking status among multicultural adolescents and provide differentiated smoking cessation education and awareness through intervention programs. This study examined the factors that influenced smoking among multicultural adolescents and identified interventions. This study was a cross-sectional survey. It used raw data from the 15th Korea Youth Risk Behavior Web-based Survey (KYRBS) of 2019. Of the 57,303 participants in the 15th KYRBS, 749 were multicultural adolescents, i.e., their fathers or mothers were not born in Korea. The factors affecting smoking among multicultural adolescents were determined using a composite sample multiple logistic regression analysis. The results of the composite sample multiple logistic regression analysis revealed that 41 (6.4%) participants were smokers, had low academic performance levels, drank alcohol, were sexually active, and were more likely to smoke with other adolescents. They were 15.79 times more likely to smoke while drinking than when not drinking. Educational and psychological interventions are needed to increase multicultural youth school adaptation and academic performance levels, reduce health risk behaviors (drinking and sexually active), and ultimately, lower the smoking rate.

## 1. Introduction

South Korea has rapidly turned into a multicultural society because of the increase in international marriages and the continued influx of foreign workers [1]. In 2019, South Korea had 137,000 multicultural students, which is a value that has tripled in the last 12 years; this number continues to increase as opposed to the decreasing total number of Korean adolescents [2]. Likewise, migrant adolescents are increasing across the world. They are either born in the host country to at least one parent born abroad or enter the host country while they are still very young [3].

Multicultural adolescents in Korea are considered to be socially and economically disadvantaged [4]. Multicultural adolescents who do not receive sufficient social support have a high level of cultural stress (awareness of discrimination, Korean language ability, or sense of alienation) [5]. Not only that, but it also affects mental health [4]. In general, adolescents feel a lot of loneliness and suffering, and there is a tendency to choose drinking and smoking as a way to overcome difficulties and instability in developmental tasks [6]. The adolescent smoking experience is highly correlated with depression, anxiety, drinking, and suicidal thoughts [7]. Given all these problems, they suffer more stress than ordinary adolescents. Similarly, they may have more risky health behaviors, such as smoking and drinking, and have a higher smoking rate [8]. Smoking among multicultural adolescents is a means of relieving suppressed emotions or a gateway to extreme choices [9]. Multicultural adolescents’ risky health behaviors vary with the composition of their multicultural families. If both parents are foreigners or if only the mother is Korean, they often smoke, drink, and engage in unsafe sexual behavior [10]. In other countries, the smoking rate of multicultural adolescents is also high, especially for girls depending on their family structure (i.e., single, foster, or no parents) [11]. Therefore, smoking should be viewed as a problem, and social attention and protection should be provided to adolescents who smoke.

In recent years, research has been conducted in Korea with an interest in multicultural adolescents, but it is still insufficient. There is no differentiated smoking prevention education program for multicultural adolescents. Therefore, this study seeks to identify the actual conditions of smoking among multicultural adolescents and the factors affecting it. The results of this study can be used as basic data for developing smoking prevention education and programs in consideration of multicultural adolescents’ characteristics. Its specific purposes were as follows: (1) to identify the participants’ general characteristics (sex, grade, parent’s education level, residence type, economic status, and academic performance level), health risk behaviors (drinking, sexually active, and drug use), mental health (stress, depression, and suicidal thoughts), and smoking; (2) to explore the differences in the level of smoking according to the participants’ general characteristics, health risk behaviors, and mental health; and (3) to identify the factors that influence their smoking.

## 2. Materials and Methods

### 2.1. Study Design

This study used a cross-sectional survey to identify the factors influencing smoking among multicultural adolescents. It used raw data from the 15th Korea Youth Risk Behavior Web-based Survey (KYRBS) of 2019.

### 2.2. Participants

Raw data from the 15th (2019) Tenth Korea Youth Risk Behavior Web-based Survey conducted annually by the Korea Centers for Disease Control and Prevention were used. A total of 57,303 people from 800 schools responded, including youth from the first year of middle school to the third year of high school. Of the 57,303 participants in the 15th KYRBS, 749 answered “no” to the question “Was your father born in Korea?” or “Was your mother born in Korea?” Therefore, these 749 respondents were our study participants. This research received Institutional Review Board approval from the Korea National Institute for Bioethics Policy (IRB No: P01-202011-21-016).

### 2.3. Tools

The study’s dependent variable was multicultural adolescent smoking, whereas the independent variables were general characteristics, health risk behaviors, and mental health related to multicultural adolescent smoking.

### 2.4. General Characteristics

The general characteristics included sex, grade, father’s education, mother’s education, residence type, economic status, and academic performance level. Sex could be answered with either “male” and “female”; grade with “middle school” or “high school”; father’s and mother’s education with “above college”, “high school”, “under the middle school”, or “unknown”; residence type with “with family” and “nonfamily”; economic status with “high”, “moderate”, or “low”; and academic performance level with “high”, “middle”, or “low”.

### 2.5. Health Risk Behaviors

Health risk behaviors included drinking, being sexually active, and drug use. Drinking status was classified as “yes” if the participants answered “yes” to the question, “Have you had more than one drink so far?” and “more than once” to the question, “In the last 30 days, about how many days did you have more than one drink?” If the participants answered “no” to both questions, their drinking status was classified as “no.” Sexually active was answered with either “yes” or “no” to the question, “Have you ever had sex?” Drug use was answered with either “yes” or “no” to the question, “Have you ever habitually or deliberately taken drugs or drank butane gas or bond so far?”

### 2.6. Mental Health

In this study, mental health included stress, depression, and suicidal thoughts. Stress was answered with “too much,” “much,” “a little,” “not much,” or “not at all” to the question, “How much stress do you usually feel?” Depression was answered with “yes” or “no” to the question, “In the last 12 months, have you felt sad or hopeless enough to stop your everyday life for 2 weeks?” Suicidal thought was answered with “yes” or “no” to the question, “In the last 12 months, have you thought about suicide?”

### 2.7. Smoking

Smoking status was classified as “yes” if the participants answered with “yes” to the question, “Have you ever taken one or two drags from a cigarette?” and “more than once” to the question, “In the last 30 days, on how many days did you smoke at least one cigarette?” If the participants answered “no” to both questions, their smoking status was classified as “no”.

### 2.8. Data Analysis

The data (15th KYRBS of 2019) used in this study were evaluated using composite sample analysis that reflected strata, cluster, weight (*W*), and a finite population modification coefficient. Data were analyzed using IBM SPSS Statistics 25. The following were analyzed:(1)The frequency and percentage of general characteristics, health risk behaviors, mental health, and smoking of multicultural adolescents.(2)The differences in smoking according to general characteristics, health risk behaviors, and mental health among multicultural adolescents by using the Rao–Scott chi-square test.(3)The factors affecting smoking in multicultural adolescents using multiple logistic regression analysis.

## 3. Results

### 3.1. General Characteristics, Health Risk Behaviors, Mental Health, and Smoking of Multicultural Adolescents

As shown in Table 1, the study population consisted of more females than males (391 [50.7%] vs. 358 [49.3%]). There were twice as many middle school students than high school students (523 [63.7%] vs. 226 [36.3%]). Regarding their fathers’ education level, the majority answered “unknown,” followed by “high school,” “above college,” and then “under middle school” (258 [33.8%], 215 [30.2%], 156 [23.8%], and 86 [12.1%], respectively). For mothers, most of the participants answered “unknown”, followed by “above college,” “high school,” and “under middle school” (301 [37.3%], 219 [30.9%], 160 [22.9%], and 65 [8.8%], respectively). Furthermore, 704 (93.1%) lived with their families, whereas 45 (6.9%) did not. The majority had a moderate economic status (381 [47.9%]), followed by high (192 [27.1%]) and low (176 [25.0%]). However, the academic performance level was mostly low among the participants (304 [42.1%]), whereas those with high and medium achievement rates were similar in number (222 [31.4%] and 223 [26.5%], respectively). Additionally, 207 (30.0%) had experienced alcohol drinking, 51 (8.2%) had been sexually active, and 19 (3.2%) had used prohibited drugs. Most of the participants perceived their stress level to be “a little” (283 [36.7%]), followed by “much” (203 [27.8%]), “not much” (130 [16.7%]), “too much” (106 [15.5%]), and “not at all” (27 [3.3%]). Meanwhile, only 205 (29.9%) were depressed and 116 (16.3%) were suicidal.

### 3.2. Differences in Smoking According to Multicultural Adolescents’ General Characteristics, Health Risk Behaviors, and Mental Health

Grade (modified F [Fm] = 9.65, *p* = 0.002), residence type (Fm = 6.01, *p* = 0.015), academic performance level (Fm = 7.15, *p* = 0.001), drinking (Fm = 62.04, *p* < 0.001), sexually active (Fm = 60.96, *p* < 0.001), drug use (Fm = 22.73, *p* < 0.001), stress (Fm = 5.29, *p* < 0.001), and suicidal thoughts (Fm = 4.82, *p* = 0.029) were significant.

The smoking experience was higher in high school students (10.9%) than in middle school students (3.8%), in participants who answered “nonfamily” (17.1%) than in those “with family” (5.6%), in participants with low school achievement (10.9%) than in those with medium (2.1%) and high (4.0%), and in participants with drinking experience (18.9%) than in those without (1.1%). Smoking was also more prevalent in sexually active (34.0%) than not (3.9%), and drug use (34.2%) than not (5.5%). The rate of smoking was 16.4% when the stress was “too much” and 3.4% when “not at all.” Lastly, the smoking experience was higher in people with suicidal thoughts (11.7%) than in those without (5.4%). Table 1 presents such results.

### 3.3. Factors Influencing Smoking among Multicultural Adolescents

We considered the statistically significant variables of general characteristics, health risk behaviors, and mental health to determine the factors influencing smoking in multicultural adolescents. We analyzed these factors using multiple logistic regression and found that the resulting model was significant (Wald F = 5.211, *p* < 0.001). The explanatory power was 41.1%. School achievement and health risk behaviors, including drinking and being sexually active, were the factors influencing smoking among multicultural adolescents.

The risk of smoking was 2.72 (95% confidence level [CI] = 1.09–6.81) times higher for those with low school achievement, 15.79 (95% CI = 5.08–49.03) times higher for those with drinking experience, and 7.44 (95% CI = 1.92–28.89) times higher for those who were sexually active. These results are shown in Table 2.

## 4. Discussion

This study used the 15th KYRBS to determine the factors that influenced smoking among multicultural adolescents in South Korea. Of the 57,303 survey participants, 749 reported that their fathers or mothers were not born in Korea; hence, they became our study participants. We found that 41 (6.4%) of the 749 participants smoked. These multicultural adolescents were more likely to smoke when they had low academic performance levels, drank alcohol, and were sexually active. 

In this study, the number of multicultural adolescents who drink was 207 (30%). Drinking was a strong predictor of current smoking status. Adolescents from multicultural families are relatively easily exposed to psychological stress due to complex problems such as cultural conflicts and different appearances [12]. To resolve these negative emotions, adolescents from multicultural families often drink alcohol [13]. Among them, 36 (18.9%) were drinking and smoking altogether. The relationship between drinking and smoking can be explained by problem behavior theory. In other words, there are several common factors in the occurrence of various risky health behaviors, and in this respect, drinking and smoking coexist [14]. For example, adolescents who drink are three times more likely to become smokers than adolescents who do not drink, and similarly, adolescents who smoke are three times more likely to become drinkers than adolescents who do not smoke [15].

According to a survey, the lifetime smoking experience rate of adolescents from North Korean families was more than twice that of South Korean adolescents, and stress was the main causative factor because of the mental and physical trauma experiences from their North Korean defection [16]. Generally, adolescents who feel stressed have a relatively high smoking rate [17]. Although stress was not a factor influencing smoking in this study, more than half of adolescents appeared to experience stress in general characteristics. Thus, in terms of the fact that stress acts as a cause of drinking [18], it is thought that stress indirectly will affect smoking through drinking. Since the ages at which drinking and smoking begins among adolescents are starting to decrease gradually and the use of various types of cigarettes such as e-cigarettes among adolescents has increased recently [19], education to prevent drinking and smoking, which threatens the health of adolescents, should be conducted early.

Adolescents who are highly sexually curious are being exposed to sex-related information indiscriminately online. Adolescent sexual experiences are problematic because they may not have acquired the right information about sex. Additionally, adolescents are still immature in regard to taking responsibility for sexually transmitted diseases and pregnancy that may occur after a sexual experience [20]. Adolescents’ sexual experience starts from curiosity and tends to occur simultaneously with drinking and smoking [21]. That is, it is related to the problem behavior of adolescents. Therefore, in developing a program to prevent smoking, proper sex education and education programs on drinking should be conducted at the same time.

Lee et al. [22] reported a high smoking rate among multicultural adolescents who had low academic performance levels, consistent with our study results. Studies on smoking among ordinary adolescents also reported a high smoking rate among adolescents with low academic performance levels [23]. According to a study by Yang et al. [1], children from multicultural families often drop out of school because of insufficient educational support, discrimination, and peer relationship problems. Ko [24] mentioned that multicultural adolescents are vulnerable to stress because of language problems, a conflict between foreign-born parents and peer groups, and discrimination in school and that they are likely to drink and smoke to avoid stress. Thus, multicultural adolescents need psychological adaptation, academic motivation improvement, and psychological counseling programs, suggesting the need for psychological and emotional development and physical support to help multicultural adolescents get along with their peers harmoniously.

Habits formed during the adolescent years affect adolescents’ future adult lives. Thus, having good habits is important to improve their health. Programs such as smoking prevention education for multicultural adolescents should be developed. Educational interventions such as self-assertion training to refuse are needed when smoking is recommended.

Regarding study limitations, this study included large-scale raw data and self-reports. Hence, this study cannot be generalized to adolescents who currently have varying cultures, nationalities, and degrees of Korean language skills. Additionally, there was no consideration of e-cigarettes, which are increasing in use among adolescents [25]. Therefore, future research should focus on factors that affect the smoking rate of multicultural adolescents according to differences in culture, nationality, Korean language skills, smoking type, and violence.

## 5. Conclusions

There were 749 multicultural adolescents from the 15th KYRBS considered in this study. Multicultural adolescents were found to smoke more when they had lower school achievement, drank, and were sexually active. In addition, those with drinking experience were 15.79 times more likely to smoke than those non-drinking.

Therefore, educational and psychological interventions should be considered, requiring a multidimensional approach including parents, peer groups, and schoolteachers. These interventions contribute to increasing the school adaptation of multicultural adolescents, lowering their experience rates of health risk behaviors, such as drinking, being sexually active, and drug use, and ultimately, reducing their smoking rate.

## Figures and Tables

**Table 1 ijerph-19-11219-t001:** Differences in smoking according to multicultural adolescents’ general characteristics, health risk behaviors, and mental health (*N* = 749).

Characteristics	Categories	*n* (%) *	Smoking		*F* _m_ ^†^	Num. d.f. ^‡^ (Denom. d.f) ^§^	*p*
			No *n* (% *)	Yes *n* (% *)			
Sex	Male	358 (49.3)	336 (93.1)	22 (6.9)	0.21	1.00 (314)	0.647
	Female	391 (50.7)	372 (94.1)	19 (5.9)			
Grade	Middle school	523 (63.7)	507 (96.2)	16 (3.8)	9.65	1.00 (314)	0.002
	High school	226 (36.3)	201 (89.1)	25 (10.9)			
Father’s education ^|^ (*n* = 715)	≥College	156 (23.8)	144(90.9)	12(9.1)	0.78	2.88 (903.58)	0.499
	High school	215 (30.2)	204 (93.8)	11 (6.2)			
	≤Middle school	86 (12.1)	79 (93.1)	7 (6.9)			
	Unknown	258 (33.8)	249 (95.1)	9 (4.9)			
Mother’s education ^|^ (*n* = 745)	≥College	219 (30.9)	208 (93.3)	11 (6.7)	2.00	2.85 (893.55)	0.116
	High school	160 (22.9)	148 (93.4)	12 (6.6)			
	≤Middle school	65 (8.8)	58 (86.8)	7 (13.2)			
	Unknown	301 (37.3)	291(96.0)	10 (4.0)			
Residence type	With family	704 (93.1)	670 (94.4)	34 (5.6)	6.01	1.00 (314)	0.015
	Nonfamily	45 (6.9)	38 (82.9)	7 (17.1)			
Economic status	High	192 (27.1)	177 (90.3)	15 (9.7)	2.34	1.99 (625.58)	0.098
	Medium	381 (47.9)	366 (95.3)	15 (4.7)			
	Low	176 (25.0)	165 (93.8)	11 (6.2)			
School achievement	High	222 (31.4)	213 (96.0)	9 (4.0)	7.15	1.83 (575.27)	0.001
	Medium	223 (26.5)	217 (97.9)	6 (2.1)			
	Low	304 (42.1)	278 (89.1)	26 (10.9)			
Drinking	Yes	207 (30.0)	171 (81.1)	36 (18.9)	62.04	1.00 (314)	<0.001
	No	542 (70.0)	537 (98.9)	5 (1.1)			
Sexually active	Yes	51 (8.2)	32 (66.0)	19 (34.0)	60.96	1.00 (314)	<0.001
	No	698 (91.8)	676 (96.1)	22 (3.9)			
Drug use	Yes	19 (3.2)	12 (65.8)	7 (34.2)	22.73	1.00 (314)	<0.001
	No	730 (96.8)	696 (94.5)	34 (5.5)			
Stress	Too much	106 (15.5)	90 (83.6)	16 (16.4)	5.29	3.71 (1164.40)	<0.001
	Much	203 (27.8)	197 (96.1)	6 (3.9)			
	A little	283 (36.7)	268 (93.9)	15 (6.1)			
	Not much	130 (16.7)	127 (97.3)	3 (2.7)			
	Not at all	27 (3.3)	26 (96.6)	1 (3.4)			
Depression	Yes	205 (29.9)	189 (91.6)	16 (8.4)	1.46	1.00 (314)	0.228
	No	544 (70.1)	519 (94.4)	25 (5.6)			
Suicidal thought	Yes	116 (16.3)	104 (88.3)	12 (11.7)	4.82	1 (314)	0.029
	No	633 (83.7)	604 (94.6)	29 (5.4)			
Total		749 (100)	708 (93.6)	41 (6.4)			

* *n* is the unweighted sample size and percent (%) is weighted percent, which is calculated using complex sample analysis; ^†^ modified *F*, calculated using complex sample analysis; ^‡^ numerator: degrees of freedom; ^§^ denominator: degrees of freedom; ^|^ skipped responses were excluded.

**Table 2 ijerph-19-11219-t002:** Factors influencing smoking among multicultural adolescents (*N* = 749).

Characteristics	Categories	*B*	OR	95% CI	*p*
Grade (ref. middle school)	High school	0.13	1.14	0.45–2.91	0.786
School achievement (ref. high)	Medium	−0.28	0.76	0.24–2.35	0.627
	Low	1.00	2.72	1.09–6.81	0.033
Residence type (ref. nonfamily)	With family	0.51	1.66	0.24–11.47	0.606
Drinking (ref. no)	Yes	2.76	15.79	5.08–49.03	<0.001
Sexually active (ref. no)	Yes	2.01	7.44	1.92–28.89	0.004
Drug use (ref. no)	Yes	0.29	1.33	0.17–10.28	0.782
Stress (ref. not at all)	Too much	1.77	5.84	0.53–64.67	0.150
	Much	0.53	1.70	0.16–18.14	0.661
	A little	1.15	3.16	0.32–31.18	0.325
	Not much	−0.45	0.64	0.05–7.58	0.722
Suicidal thought (ref. yes)	No	0.37	1.45	0.46–4.52	0.525
Nagelkerke *R*^2^ = 0.411, Cox and Snell *R^2^* = 0.156, Wald *F* = 5.211, *p* < 0.001					

CI, confidence interval; OR, odds ratio; ref., reference.

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
