# Peer review of "Factors Influencing Smoking among Multicultural Adolescents"

_ijerph, 2022, doi:10.3390/ijerph191811219_

Round 1

Reviewer 1 Report (Previous Reviewer 2)

The authors have adequately responded to the comments from previous reviews, and the quality of the manuscript has significantly improved. Still, there are some minor errors that need to be addressed.

1. Abstract, correct “Had sexually active” for “were sexually active.”

2.- Introduction (2nd paragraph) “And adolescent smoking experience…”  eliminate the AND and start the sentence from “Adolescent smoking…”.

3.- The authors corrected the term to “sexually active,” as suggested. However, a revision must be made to be sure that the correct grammatical form of the term is being used. E.g.,”…they often smoke, drink and engage in unsafe sexually active” needs to be “… engage in unsafe sexual behavior” (Introduction 2nd paragraph).

4.- If possible, I suggest the authors add, in the Methods section, the ranges of wealth used for the “economic status “classification into low, moderate and high. Also, if the quantity is in South Korean won, provide their equivalent in USD.

5.- Correct the capital letter in “low School achievement” (Results, section 3.3)

6.- Discussion (2nd paragraph) Please explain what do you mean by “drinking affecting smoking” and how that is different from “drinking and smoking altogether”?

7. Finally, I suggest the authors check the congruency of the terms used in the paper.

Author Response

Thank you for your review, which have helped us improve the quality of this manuscript. According to the reviewers’ comments, we have made revision on the manuscript. We used red colored fonts to indicate the revised parts for your recognition. 

Reviewer 2 Report (Previous Reviewer 3)

The proposal of change are accepted and the paper now is fine

Author Response

Thank you for your, which have helped us improve the quality of this manuscript. 

This manuscript is a resubmission of an earlier submission. The following is a list of the peer review reports and author responses from that submission.

Round 1

Reviewer 1 Report

CENTRAL AND GENERAL ISSUES

Summary

This study examines the factors that influence smoking among multicultural adolescents and identifies interventions. I believe that the findings are relevant and add value to the literature.. However, I think there are some important aspects that need to be improved before recommending its publication in Adolescents.

Specific Comments

1. The introductory section does not explicitly talk about how tobacco companies are using electronic devices and heated tobacco to replace traditional cigarettes with these new alternatives. This issue is important because heated tobacco is being used as an alternative in many countries and, in many cases, the market share of these products is particularly high. A recent article shows how Philip Morris International is using heated tobacco to replace the traditional cigarette. In this line, I think the authors should reflect on this and cite the following paper:

Golpe, A. A., Martín-Álvarez, J. M., Galiano, A., & Asensio, E. (2022). Effect of IQOS introduction on Philip Morris International cigarette sales in Spain: a Logarithmic Mean Divisa Index decomposition approach. Gaceta Sanitaria. https://doi.org/10.1016/j.gaceta.2021.12.007

2. The authors discuss in the introduction many variables that influence adolescents' choice to smoke. However, they do not talk about the exposure of these adolescents to the marketing actions of the companies and how this can influence their decision to smoke. There is a recent study that indicates that the regulation of the tobacco market causes a homogenization of the diffusion of products, due to the impossibility of differentiating through innovation. In addition, other equally recent work indicates that regulation of packaging-related features will affect CCL choices among young adult smokers in the US. I believe those two articles should be cited, the references of which are:  

Almeida, A., Galiano, A., Golpe, A. A., & Martín Álvarez, J. M. (2021). The Usefulness of Marketing Strategies in a Regulated Market: Evidence from the Spanish Tobacco Market. E&M Economics and Management, 24(2), 171–188. https://doi.org/10.15240/tul/001/2021-2-011

Shang, C., Nonnemaker, J., Sterling, K., Sobolewski, J., & Weaver, S. R. (2021). Impact of Little Cigars and Cigarillos Packaging Features on Product Preference. International Journal of Environmental Research and Public Health, 18(21), 11443. https://doi.org/10.3390/ijerph182111443

3. In the conclusions I see that they have concluded about the findings found and have marked lines of future research. However, I do not see any paragraph showing the limitations of this work. It would be important for the limitations of this paper to be made clear.

Author Response

I have modified it based on what you said.
And I have corrected all of the discussion.

Reviewer 2 Report

Title: Factors Influencing Smoking Among Multicultural Adolescents.

Manuscript ID: Adolescents-1826210

General Comments: Thank you for the opportunity to review this manuscript. The authors cover the relevant topic of characterizing the factors influencing cigarette use in multicultural adolescents living in South Korea. There has been a rapid increase in the number of multicultural families in Korea in a country where monocultural families were the standard. This paper highlights the importance of the stressors that multicultural adolescents are exposed to in their health risk behaviors, like smoking. There is value in the article. However, there are some comments, if addressed, I believe will strengthen the quality of this commendable paper.

1. Introduction. I suggest the authors include the prevalence of multicultural families and multicultural adolescents in South Korea. This will help the international reader to contextualize the size of the problem and the relevance of addressing it.

2. The study population in Reference 4 in the Introductory text refers to a population from Turkey. Cultural norms might vary significantly within countries. I suggest strengthening this argument by including a reference related to the South Korean population. (e.g., Social Determinants of Health and Well-Being of Adolescents in Multicultural Families in South Korea: Social-Cultural and Community Influence, Shin 2021 & Social Determinants of Health of Multicultural Adolescents in South Korea: An Integrated Literature Review (2018~2020) Kim, Youlim, 2021).   

3. Last paragraph of the Introduction section. The authors state that “The results of this study can be used as basic data for developing smoking prevention education programs” This is an exciting point that was not revisited in the Discussion section. I suggest the authors add this in the Discussion section by mentioning which smoking educational programs for adolescents are in place and which changes, based on the results, could be made to adapt them to multicultural adolescents.

4. Methods Section. I strongly suggest the authors include a brief description of the 15th Korean Youth Risk Behavior survey design and data collection methods, including the population’s age range and a reference where the reader can find detailed information about the survey.

5. The authors do not state the selection process of the independent variables. Please include a brief description of the process. The paper by Ahn IY, 2016 (Ref 5) mentions a potential association between smoking and experiencing violence. Why do the authors not consider this risk factor for their analysis?

6. Why do the authors not consider including “frequency of use” as part of their cigarette user definition? It is well known that “frequency of cigarette use” is associated with other high-risk behaviors. The analytical sample includes experimenters and regular users with their current smoker definition. I suggest addressing this within the limitation sections.

7. In addition, e-cigarette use has been increasing among Korean adolescents. In the US, it has been observed that initiation with e-cigarettes can lead to cigarette consumption later in life. Is there a particular reason why the authors do not consider including e-cigarettes or consumption of other tobacco products for their analysis?  

8. Results. First paragraph. “whereas those with high and low achievement rates were similar in number (222 [31.4%] and 223 [26.5%], respectively),” please correct the low to medium.

9. Correct for double parenthesis in Table 1. Sex Female (5.9))

10. Results sections 3.3. I suggest the authors include in the description of the results the comparison group when referring to “increased risk.” In addition, I propose reporting results in the format of OR (95% CI); there is no need to include the B and P values.

11. Please consider changing the term “sexual activity” to “sexually active” in the text. It translates better into English when referring to individuals who have engaged in sexual activity.

12. Discussion. Please consider rephrasing the “...15.79 times higher when they were drinking” to “multicultural adolescents who smoke were more likely to report alcohol consumption than non-cigarette users.”

13. Discussion, 2nd paragraph. The evidence used by the authors to support their results of “multicultural adolescents who are drinking are more likely to smoke” may not adequately support their results. The evidence used only presents the prevalence of smoking and drinking in adolescents separately but not the % of adolescents who used cigarettes and consume alcohol or evidence of an association.   

14. Discussion 3rd paragraph. The authors concluded here that the present study supports the evidence from other studies that individuals who smoke are more likely to consume drugs. However, the results reported in Table 2 do not support that conclusion. Please explain and address the possible cause of the discrepancy between your evidence and the published evidence. Something similar is seen with the lack of association between stress levels and smoking in this study, contrasting the presented evidence. Please addressed.  

15. The authors develop a fascinating statement regarding the influence of second-hand smokers (family and peers) and how It influences adolescents’ smoking behavior. Although interesting, this topic may be beyond the scope of this study as the authors do not evaluate that variable within their analysis. I suggest rephrasing this statement within the text to highlight its relevance within this study.

16. The sample size of multicultural adolescents who use cigarettes in the study is an important limitation that has potentially influenced the lack of associations found in their results. Please consider addressing this and the cross-sectional design of the survey within the limitation of your study.

17. It would be exciting and could strengthen their argument of the importance of developing programs for multicultural adolescents to show how the characteristics that influence smoking in multicultural adolescents differ from those of monocultural adolescents. Please consider

Author Response

(The authors gave the same response as above.)

Reviewer 3 Report

The paper is quite interesting survey coordinated by Korean Ministry. The present data are collected in 2019 even if the survey with data of 2011-2016  was already pubished.

This paper so doesn't add anything of new o the previous research.

The authors should try to explain the interested process with the help of different variable more connected to the recent literature. One of the main result, the link between the low academic performance and the fact that parents are not born in Korea is nt well described.

In fact the parents non Korean in some case was in Korea from their first year of life so they know very well the native language.

It's appropriate so to find other variable involved in the phenomenon under analysis.

Author Response

(The authors gave the same response as above.)

Reviewer 4 Report

The article Factors Influencing Smoking among Multicultural Adolescents describes the analysis of data from the 15th Korea Youth Risk Behavior Web-based Survey (KYRBS; 2019) with three goals:

1. Understanding the general characteristics of multicultural youth living in Korea
2. Examine the correlation between those general characteristics and smoking status
3. Identify factors that predict smoking status

The goals are well-outlined and the paper sets a clear intention. The statistical design is well-planned and the results are clearly presented. The paper can be improved through thoughtful consideration of the outline and additional information presented in the introduction. Specific comments below:

Abstract: It is clear that the paper is interested in multicultural adolescents but foundational information about why those adolescents are known to be different from their Korean counterparts is unclear. Some information about that up front would add a lot of weight to the paper and your results. Similarly, at the end of the abstract, the authors mention the need for educational and psychological interventions. Do those exist for Korean adolescents but were found to be ineffective for multicultural adolescents? The element I felt was missing was a comparator - why should the reader suspect that this group is different from others and in need of special consideration. I'm sure the literature is available and don't doubt the legitimacy of the need to study this population, but it would help my understanding to have that presented as early as possible in the paper.

Introduction: Be cautious with assertions that individuals who are multicultural are fundamentally different or deviant. From a lens of intersectionality, there are a multitude of factors that could influence risky health behaviors and we, as scientists, should be cautious about suggesting that characteristics of an individual are inherently bad or likely to lead to negative behaviors.

I would avoid using the phrase "deviant" throughout as that has a highly negative connotation that I don't believe is the intention. Consider "risky behaviors" or "risky health behaviors" or something similar. Along those lines, toward the middle of the second paragraph of the introduction you say "Multicultural teenagers' deviant behaviors vary by..." - maybe consider language such as "Factors that influence the likelihood to engage in risky health behaviors include family composition."

Just after citation 8: Are those experiences unique to multicultural teenagers or is this similar for all adolescents who smoke? I am left wanting for a comparator so I know how important the information is in the larger context. Would also avoid using the phrase "go along with" - does this imply a correlation? Is there a causal mechanism involved?

Final paragraph of introduction: "...the actual conditions of smoking..." - could this be elaborated? Is this age at onset, smoking device, etc?

Final paragraph of introduction: The reader is left with the notion that the results of the study will assist in developing cessation and prevention materials that are specific to multicultural youth but without knowing how multicultural youth are different from the larger population, it is not clear that these materials would not be good for others. 

Overall: The introduction would benefit from some fresh organization. Information about how multicultural Korean adolescents are different from single-culture Korean adolescents and why those differences pose a problem for the effectiveness of smoking prevention/cessation programs is crucial.

Participants: A description about why there was no comparison group might be helpful here. I realize it is illogical to ask for a statistical comparison post hoc here but perhaps information in the introduction about how so much is already known about that sample (and the information that is known) would relieve this need?

Smoking: Does the survey inquire about other forms of smoking? In the US, we are seeing rates of adolescent cigarette use drop dramatically but rates of use are elevated for electronic nicotine delivery systems and hookah, for example. If the data is not available, a note on that would be useful.

Results, section 3.2: The authors note that more high school students are smoking than middle school students - is age at initiation accounted for here? We would expect that middle school students who become high school students continue to smoke, perhaps, so the differential might not be as profoundly meaningful.

Throughout, try to avoid using the word "during" (e.g., "Smoking was also more prevalent during sexual activity than not...") as that implies co-occurrence of use (i.e., smoking at the same time one is engaged in sexual activity).

Is the difference in smoking rates when adolescents were stressed "too much" statistically different from the rates when people were "not at all" stressed? When more than one category is presented for a variable, consider using an asterisk or other to denote the groups that are significantly different.

Table 2: Though I realize it does not make a statistical difference, perhaps consider making "With family" and "No suicidal thoughts" the reference groups. For all other categories the authors seem to have chosen the normative and larger groups as the reference, which is a very sensible choice and lends clarity to the findings.

Discussion: Similar to the comment regarding the use of the phrase "during," here the authors state that "...was 15.79 times higher when they were drinking." Maybe something like "among those who also reported alcohol use"

3rd paragraph: The authors describe a study that seems to have investigated multicultural youth in relation to a control group of youth whose parents are both Korean. It is not fair to assume that this study had the same findings as there was no comparator. Consider removing this altogether. 

The same paragraph states "These habits will lead them to deviate problems, such as violence and having sexual activity." This should be referenced and also stated in a less factual way, perhaps something along the lines of "These habits are highly correlated with violence and increased risky sexual behavior," depending on the information provided in the reference.

The same paragraph ends with information about the age of smoking initiation which was not reviewed in this paper and should be excluded.

5th paragraph: While the authors make a strong and clear point about increased stress, this paragraph doesn't seem to have a strong place in the discussion, perhaps consider rewording to bring the conversation back to the results presented.

8th paragraph: Unclear what is meant by the phrase "Similarly, the more neglect and abuse they experience from their parents, the more influence their deviant behaviors have."

9th paragraph: The opening statements are a little awkward to read, consider revising for clarity.

The end statement, assuming that students with high academic performance smoke less because they have lower stress levels, high mental health and high psychological wellbeing seems to be a stretch. The results do not support this contention and the statement is not referenced.

Paragraph 11: This study does not have sufficient information to make statements about the "fear of being excluded or ostracized by their peers when they refuse to smoke." This paragraph in general does not seem backed by the evidence presented in the study and can likely be excluded.

The discussion, in general, is very long and not always related directly to the findings. Some information may be best presented in the introduction to describe and validate your chosen methodology (esp. why the authors selected certain independent variables). The authors should avoid describing constructs that were not included in the analysis and making assumptions using constructs that were outside the scope of this analysis.

Conclusion: In the first sentence, perhaps consider giving the percentage of smokers (6.4%) and then, in a separate sentence, the characteristics of those smokers. As it reads now, it seems like 6.4% of individuals were smokers, had poor academic performance, drank alcohol, and engaged in sexual activities.

Author Response

(The authors gave the same response as above.)

Round 2

Reviewer 4 Report

Thank you for taking the time to consider my comments when editing the document. I approve the manuscript in its present condition.